# Contribution of the α5 nAChR Subunit and α5SNP to Nicotine-Induced Proliferation and Migration of Human Cancer Cells

**DOI:** 10.3390/cells12152000

**Published:** 2023-08-04

**Authors:** Irida Papapostolou, Daniela Ross-Kaschitza, Florian Bochen, Christine Peinelt, Maria Constanza Maldifassi

**Affiliations:** Institute of Biochemistry and Molecular Medicine, University of Bern, 3012 Bern, Switzerland; irida.papapostolou@unibe.ch (I.P.); daniela.ross@unibe.ch (D.R.-K.); florian.bochen@unibe.ch (F.B.); christine.peinelt@unibe.ch (C.P.)

**Keywords:** cancer, proliferation, migration, nicotinic acetylcholine receptors, α5 nicotinic receptor subunit, nicotine, ion channel

## Abstract

Nicotine in tobacco is known to induce tumor-promoting effects and cause chemotherapy resistance through the activation of nicotinic acetylcholine receptors (nAChRs). Many studies have associated the α5 nicotinic receptor subunit (α5), and a specific polymorphism in this subunit, with (i) nicotine administration, (ii) nicotine dependence, and (iii) lung cancer. The α5 gene *CHRNA5* mRNA is upregulated in several types of cancer, including lung, prostate, colorectal, and stomach cancer, and cancer severity is correlated with smoking. In this study, we investigate the contribution of α5 in the nicotine-induced cancer hallmark functions proliferation and migration, in breast, colon, and prostate cancer cells. Nine human cell lines from different origins were used to determine nAChR subunit expression levels. Then, selected breast (MCF7), colon (SW480), and prostate (DU145) cancer cell lines were used to investigate the nicotine-induced effects mediated by α5. Using pharmacological and siRNA-based experiments, we show that α5 is essential for nicotine-induced proliferation and migration. Additionally, upon downregulation of α5, nicotine-promoted expression of EMT markers and immune regulatory proteins was impaired. Moreover, the α5 polymorphism D398N (α5SNP) caused a basal increase in proliferation and migration in the DU145 cell line, and the effect was mediated through G-protein signaling. Taken together, our results indicate that nicotine-induced cancer cell proliferation and migration are mediated via α5, adding to the characterization of α5 as a putative therapeutical target.

## 1. Introduction

Nicotinic acetylcholine receptors (nAChRs) are pentameric ligand-gated channels that, once activated, are permeable to sodium (Na^+^), potassium (K^+^), and calcium (Ca^2+^) ions [1,2,3]. In mammals, 16 different subunits have been identified [4]. Similarly to other pentameric receptors, such as γ-Aminobutyric acid type A (GABA-A) receptors [5,6], nAChRs can assemble as either homomeric or heteromeric channels, whereby the subunit composition determines the channel’s kinetic and pharmacological characteristics [7,8,9]. Although nAChRs are known to be widely expressed through the peripheral nervous system (PNS) and central nervous system (CNS) [10], they are also found throughout the entire body, where they mediate diverse functions [11,12]. It is now recognized that apart from their very well-described ionotropic properties, these receptors also have metabotropic characteristics and activate several intracellular signaling pathways, not only in neurons but also in various other cell types [13,14,15]. Endogenous agonists, such as acetylcholine, and exogenous agonists, such as nicotine, activate these receptors by binding to very well-characterized sites found between subunits [16,17].

In the context of lung cancer, nicotine exposure is known to increase cell proliferation, cell invasion, migration, Ca^2+^ influx, epithelial to mesenchymal transition (EMT), and to initiate specific signaling cascades because of nicotine’s interaction with nAChRs [18,19,20,21,22,23]. Nicotine can also modify the expression of immune-regulatory proteins such as CD47 and PDL1 [24,25,26,27,28,29]. Several researchers have associated the nAChR subunits α5, α3, and β4 with lung cancer [30,31,32]. Additionally, in lung-cancer-derived cell lines, the activation of α5 induces proliferation, migration, invasion, and EMT transition events [33,34,35]. Interestingly, α5 fails to contribute to agonist binding but acts as an accessory subunit [36,37,38], where α5 can influence receptor traffic, sensitivity, efficacy, and Ca^2+^ permeability, such as in the α4β2α5 and α3β4α5 receptors [39]. To date, however, how α5 modulates the signaling properties of nAChRs in cancer remains an unresolved question [40]. Notably, the expression of receptors containing α5 in the brain is important for dopamine release and attention [41,42] and is involved in nicotine’s administration and withdrawal effects [43]. Consequently, α5-containing nAChRs are putative drug targets because of the association between lung cancer risk and nicotine’s addictive effects.

Further studies have shown that a specific polymorphism in this subunit, which changes an aspartate, D, in position 398 to an asparagine, N, (referred to as α5SNP), is strongly linked with nicotine dependence, quantity of smoking, and lung cancer [44,45,46,47]. Several reports claim that patients who carry the α5SNP are more prone to chronic pulmonary disease (COPD) independent of their smoking status [48,49]. Humanized mouse models expressing α5SNP confirm these data and associate this polymorphism with altered airway epithelial remodeling, as well as with an increase in basal cellular proliferation [32]. In epithelial lung cells, α5SNP was found to cause an increase in proliferation dependent on adenylyl cyclase (AC) activation [32]. Furthermore, this α5SNP is located at the intracellular loop localized between the M3–M4 transmembrane domains, an area known to be relevant for G_o_–protein interactions [50] but with a largely unknown structure [38]. Likewise, it was proposed that α5 modulates the recovery from receptor desensitization, with its expression resulting in a higher rate of receptor signaling [40].

Apart from lung cancer, current data also associate cigarette smoking with other types of cancer, such as prostate cancer (PCa) [51,52], colorectal cancer (CRC) [53], and breast cancer [54,55]. Several reports show the expression of diverse nAChR subunits in these cancer types and indicate that the activation of nAChRs stimulates cellular migration and EMT [56,57]. To date, the relevance of α5 in these nicotine-induced effects is completely unknown. In the present study, we provide information on the function of α5 in human cancer cell lines of different origins and suggest a general role of the α5 nicotinic receptor subunit in cancer cell migration and proliferation. Using datasets from The Cancer Genome Atlas (TCGA) and qPCR, we found that among nAChR subunits, α5 was prominently expressed throughout different types of cancer. Upon pharmacological blocking of nAChRs or siRNA-based silencing of α5, the nicotine-induced proliferation, migration, and EMT transition in breast, prostate, and colon cancer cells were reduced. In addition, the expression of α5SNP can enhance basal cell proliferation and migration in DU145 PCa cells, independent of nicotine exposure. In summary, our findings present valuable insights into human α5 signaling in nicotine-induced migration and proliferation, underlining α5′s role as a future therapeutical target.

## 2. Materials and Methods

### 2.1. Cell Culture

The following human cancer cell lines were used: MCF7 (breast cancer; MEM; Gibco, Waltham, MA, USA); A549 (lung cancer; DMEM; Gibco, Waltham, MA, USA); PC9 (lung cancer; RPMI; Gibco, Waltham, MA, USA); SW480 (colorectal cancer; RMPI; Gibco, Waltham, MA, USA); SW620 (colorectal cancer; L15; Gibco, Waltham, MA, USA); CACO2 (colorectal cancer; DMEM; Gibco, Waltham, MA, USA); DU145 (prostate cancer; DMEM; Gibco, Waltham, MA, USA); PC3 (prostate cancer; RMPI; Gibco, Waltham, MA, USA); and LNCaP (prostate cancer; RMPI; Gibco, Waltham, MA, USA). All cell lines were obtained from ATCC (Rockville, MD, USA). The corresponding media for each cell line was supplemented with 10% FBS (Gibco, Waltham, MA, USA), and additionally supplemented with 2 mM L-glutamine (Gibco, Waltham, MA, USA), 1% NEAA (Gibco, Waltham, MA, USA) in the case of DU145. Cells were grown in a humidified atmosphere containing 5% CO_2_ at 37 °C.

### 2.2. Drug Treatment

Nicotine and three nAChR antagonists (the non-specific inhibitor mecamylamine MEC, an α7 receptor inhibitor methyllycaconitine MLA, and a β2 subunit containing receptor inhibitor dihydro-β-erythroidine DHBE), together with the adenylyl cyclase inhibitor SQ22536 and the G_αi_ signaling blocker pertussis toxin PTX, were evaluated in cellular proliferation and migration assays. Additionally, their ability to induce changes in the expression of certain proteins was assayed via qPCR. All antagonists and agonists were obtained from Tocris (Bristol, UK), and were dissolved in H_2_O according to the manufacturer’s instructions. When antagonists’ effects were evaluated, these effects were co-applied with nicotine or alone for the control experiments.

### 2.3. Proliferation Assay

Cell proliferation was evaluated as previously published by our group [58,59] using xCELLigence^®^ E-Plates (ACEA Biosciences, San Diego, CA, USA). This system monitors changes in electrical impedance and records them as an increasing number of cells adhering to the surface of the plates. Depending on the cell type, 2 × 10^4^ or 4 × 10^4^ cells were plated in their corresponding complete growth medium with or without an agonist/antagonist, and proliferation was measured for 48–72 h in 15 min intervals. Statistical significance was analyzed using the Kruskal–Wallis test for non-parametric data with the GraphPad Prism (Version 9.3.1) software.

### 2.4. Quantitative Real-Time PCR (qPCR)

The expression of diverse genes was evaluated following previously published procedures from our group [58]. In brief, RNA from the diverse cell lines and treatments was obtained using a QIAshredder kit (Qiagen, Hilden, Germany) followed by an RNeasy Mini kit (Qiagen, Hilden, Germany). Afterward, reverse transcription of 2 µg RNA was performed (High-Capacity cDNA Reverse Transcription Kit, Thermo Fisher Scientific, MA, USA), and gene expression was evaluated using a TaqMan Gene Expression Assay. All determinations were performed in triplicate. Results were analyzed with the ΔCt method, and expression levels were normalized to the housekeeping gene TATA-binding protein. When comparing relative changes in protein expression in treated cells, the expression was compared to non-treated cells. Table 1 shows the ThermoFisher (Waltham, MA, USA) primers/probes used to quantify expression levels:

### 2.5. Transwell Migration Assay

Transwell migration of cells was evaluated as previously published by our group [58] using the MCF7, DU145, and SW480 cell lines. Here, 4 × 10^4^ cells were seeded in 0% serum media in the top chamber of xCELLigence^®^ CIM-Plates (ACEA Biosciences, San Diego, CA, USA). The respective culture media with 10% serum was added to the lowest chambers and served as a chemoattractant. For MCF7 cells, 1% serum was used in the top chamber. When compounds were being tested, these solutions were added to the top chamber. Afterward, migration was measured in 15 min intervals for 25 h. Statistical significance was analyzed using the Kruskal–Wallis test for non-parametric data with GraphPad Prism (Version 9.3.1) software.

### 2.6. Small Interfering RNA Transfection (siRNA)

We performed the siRNA transfections with a mix of 4 different human α5 siRNAs using a 1:1:1:1 ratio in a final concentration of 0.12 nmol siRNA. All siRNAs were obtained from Qiagen. The α5 siRNAs used were as follows: Hs_*CHRNA5*_7, target sequence CTGGTATCCGTATGTCACTTA; Hs_*CHRNA5*_6, target sequence CTGGACTCCACCGGCAAACTA; Hs_*CHRNA5*_5, target sequence ATGGATCACAGGTTGATATAA; and Hs_*CHRNA5*_4, target sequence CTGAGTAACAGCTAATCTTTA. Control cells were transfected with non-silencing siRNA (ns-RNA) (Qiagen, Hilden, Germany), which presented no homology with any known mammalian gene. We performed the siRNA transfections using Interferin siRNA Transfection Reagent (Polyplus, Illkirch-Graffenstaden, France). Cells were incubated overnight with a transfection mix consisting of a combination of the corresponding serum-free media, the siRNA of interest or ns-RNA, and the Interferin solution. Afterward, the cells were exposed to treatments as specified in the drug treatment section, and the expression of diverse proteins was analyzed via qPCR. Treated cells were also used for proliferation and migration assays. Knockdown verification was performed 24 h or 48 h after siRNA transfection, and α5 expression was analyzed with qPCR.

### 2.7. Constructs and Nucleofection

The α5.pcDNA3.1/V5 and α5D398N.pcDNA3.1/V5 epitope-tagged constructs used for transfection were a generous gift from Dr. Larry S. Barak and have been described elsewhere [40]. DU145 cells were transfected with either plasmid using an SE Nucleofection kit (Lonza, Basel, Switzerland). Briefly, 1.5 × 10^6^ cells suspended in nucleofection SE solution were transfected with the appropriate constructs using the DU145-specific program of the nucleofector. Next, cells were seeded and incubated for an additional 24 h before starting proliferation and migration experiments. Transfection verification was performed 24 h after nucleofection, and α5 expression was analyzed via qPCR and IF.

### 2.8. Immune Fluorescence (IF)

To positively ascertain the transfection of α5.pcDNA3.1/V5 and α5D398N.pcDNA3.1/V5 epitope-tagged constructs (a generous gift from Dr. Larry S. Barak) [40] into DU145 cells, specific immunofluorescence staining was performed on fixed cells. In brief, 24 h after nucleofection, the cells were fixed using 4% PFA in PBS (ThermoFisher, Waltham, MA, USA) for 15 min, and subsequently incubated with anti-V5 antibody (1:500, Cell Signaling, Danvers, MA, USA) overnight at 4 °C. Afterwards, cells were incubated with AlexaFluor 488 (1:1000, Invitrogen, Waltham, MA, USA) for 1 h. Samples were visualized using a Nikon Eclipse Ti2-E Widefield microscope with a Photometrics BSI camera and SpectraX laser as light source, with either a 40× or 100× objective lens.

## 3. Results

### 3.1. Expression of nAChR Subunits in Diverse Cancer Cell Lines: Focusing on the α5 Subunit

First, to assess the expression of diverse nAChR subunits in various cancers, we analyzed data from the Human Protein Atlas [60], which is based on the Cancer Genome Atlas TCGA project [61]. Figure 1A,B show that throughout different types of human cancers, α5 is a prominently expressed subunit.

Next, we investigated the expression of α5 and other nAChR subunits that are known to mediate nicotine’s pro-oncogenic effects in different human cancer cell lines. Despite the fact that the maximum expression of the α5 was in urothelial and testicular cancer, our study focused on the cancer types with the highest incidence and mortality [62], namely prostate, colon, and breast cancer. Correspondingly, we used the breast cancer cell line MCF7; the lung cancer cell lines A549 and PC9; the colorectal cancer cell lines SW480, SW620, and CACO2; and the prostate cancer cell lines DU145, PC3, and LNCAP. The gene expression profiles were studied via qPCR analysis, and the results are presented in Figure 2 and Table 2. The results show that nAChR mRNAs were differentially expressed throughout the different cell lines. In a comparison between the cell lines, the α3 and α4 subunits showed the strongest expression in SW480 cells; in a comparison between the subunits α5 was the most prominently expressed subunit in all tested cell lines. The α7 and dupα7 transcripts were differentially expressed throughout all cell lines. Interestingly, α9 was barely detectable across all cells. Lastly, β2 and β4 were also expressed but differed between cell lines.

Thus, the nAChR mRNA expression profiles demonstrate that α5 is the most significantly expressed subunit in lung, breast, CRC, and PCa cancer cells, albeit at different levels. Previous reports showed that α5 is upregulated in human lung cancer tissue [19], hepatocellular carcinoma tissue [63], gastric cancer tissue [64], and prostate cancer tissue [53]. Therefore, overexpression of the α5 subunit may contribute to the general pathological role of nicotine in cancer cells.

### 3.2. Nicotine Increases Proliferation and Migration in Various Human Cancer Cell Lines through nAChRs

We next studied whether nicotine alters proliferation in several types of human cancer cell lines through the activation of nAChRs. We tested nicotine-induced proliferation in the breast cancer cell line MCF7, the CRC line SW480, and the PCa cell line DU145. For this analysis, cells were incubated with increasing concentrations of nicotine, from 0.1 μM to 10 μM, for 72 h. These concentrations were previously used to study nicotine as an inducer of proliferation in cancer cells [20,65]. To determine the nicotine-induced proliferation, we employed a label-free impedance-based xCELLigence system that was also used by our group in a previous study [59]. In this system, an increasing number of cells on a plate raises the electrical impedance, which can be displayed as an increase in the cell index parameter. Figure 3 shows that in the presence of 1 µM nicotine, proliferation was significantly increased. In the MCF7 cell line (Figure 3A,B), this effect was already significant after 12 h, whereas the effect was significant in DU145 (Figure 3G,H) and SW480 (Figure 3D,E) cells after 24 h of incubation. Nicotine thus consistently increases proliferation but to a varying extent at different time points between cell lines tested. Our results are in line with previous observations in the lung cancer cell line A549 and the breast cancer line MCF7, in which the authors observed that nicotine augmented the proliferation of both cell types with the maximum effect observed at a concentration of 1 µM nicotine [20].

To challenge the hypothesis that the increase in proliferation with 1 µM nicotine is caused by nAChR activation, proliferation was tested with nicotine in the presence of the non-specific nAChR antagonist mecamylamine (MEC). Since earlier work showed the relevance of α7 in nicotine’s pro-oncogenic effects [66], and because α7 mRNA was detected in the cell lines used (Figure 2), we also tested the role of α7 in nicotine-induced proliferation using a known α7 inhibitor (methyllycaconitine, MLA). Some cell lines also expressed detectable amounts of the β2 transcript (SW480). Thus, the β2 inhibitor dihydro-beta-erythroidine (DHBE) was also employed. As outlined above, we used an impedance-based label-free xCELLigence assay and induced proliferation using 1 μM nicotine with or without 10 μM MEC, 10 nM MLA, or 10 μM DHBE. Previously, these concentrations were shown to inhibit the corresponding nAChR subtypes in different neuronal cell types [67]. As presented in Figure 3C,F,I, nicotine induced an increase in proliferation, reflected by an increase in the cell index. In the presence of MEC, nicotine failed to increase proliferation in all tested cell lines, demonstrating that nicotine’s effect was mediated through activation of nAChRs. On the other hand, the presence of MLA or DHBE failed to inhibit nicotine-induced proliferation, showing that α7 and β2 are not involved in nicotine-induced signaling. When applied without nicotine, the inhibitors did not modify the basal proliferation of SW480 or DU145 cells, demonstrating that they do not interfere with normal cellular growth mechanisms (Appendix A).

Additionally, we evaluated whether nicotine, through nAChRs, affected migration in the DU145 cell line. Migration was investigated using an impedance-based assay with FBS as a chemoattractant [68]. As shown in Figure 4, when cells were incubated with 1 μM or 10 μM nicotine, we observed augmented cell migration at 24 h. This effect was significant for 1 μM nicotine. As described above, we used the non-specific nAChR antagonist MEC to determine the contribution of nAChRs to this effect. Additionally, since it was shown that α7 is important for nicotine-induced migration in the A549 cell line [20], and because the qPCR analyses indicated that α7 is also expressed in DU145 cells, we also used the α7 receptor inhibitor MLA. The β2 inhibitor DHBE was evaluated as well. Figure 4 shows that in the presence of MEC, migration was not affected by nicotine, reflecting the clear contribution of nAChRs to nicotine-induced migration. MLA failed to affect migration, and DHBE also impaired nicotine’s effects.

When migration was studied in the SW480 cells (Appendix A), a more constrained migration was observed. Here, exposure to nicotine augmented the migration index, but because of the limited migration, no further studies were performed on these cells. In the MCF7 cell line, nicotine failed to induce migration at both tested concentrations (Appendix A).

### 3.3. siRNA Based Silencing of the α5 Subunit Reduces Nicotine-Induced Proliferation and Migration in Several Cancer Cell Lines

As our qPCR studies indicated high expression levels of α5 in all cell lines, we sought to determine whether nicotine could modify proliferation and migration through the activation of α5-containing nAChRs in the analyzed cancer cell lines. As no specific agonist exists for α5-containing receptors [69], we studied the relevance of α5 by silencing the subunit using specific siRNAs. MCF7, SW480, and DU145 cells were transfected either with a mixture of four siRNAs specifically directed towards the α5 subunit (α5-siRNA) or with a non-silencing control RNA (ns-RNA). Proliferation and migration were also evaluated. The qPCR analysis in Figure 5 shows that siRNA-based knockdown of the α5 gene *CHRNA5* caused a ~90% decrease in transcripts in MCF7 and SW480 cells and a ~70% decrease in the DU145 cell line when evaluated at 24 h (Figure 5A) and 48 h (Figure 5B) post-transfection. As a control, we also determined the expression levels of other nAChR subunits (Appendix A), and no considerable changes in expression levels were detected.

Next, we analyzed the contribution of α5 in the nicotine-induced proliferation in MCF7, SW480, and DU145 cells. Cells were transfected with α5 siRNA or ns-RNA, and proliferation was detected as described above. Figure 5C–E show that 1 μM nicotine increased proliferation in ns-RNA-transfected cells, which is comparable to the nicotine-induced increase in proliferation in non-transfected cells. In cells transfected with α5 siRNA, nicotine failed to stimulate proliferation. When nicotine was absent, no statistically significant difference between ns-RNA- and α5-siRNA-transfected cells could be observed. This result indicates that in these cells, the presence of this subunit is not relevant for cellular proliferation in the absence of a nAChR agonist. Additionally, nicotine-induced proliferation is mediated via nAChRs, with the presence of α5 being indispensable.

In the previous sections, we showed that nAChRs mediate nicotine-induced migration. Subsequently, we tested if this effect is specific to α5-containing receptors. DU145 cells were transfected with α5-siRNA or ns-RNA, and after 24 h, migration was determined as described above. As shown in Figure 5F, 1 μM nicotine promoted migration in ns-RNA-transfected cells at 24 h, an effect comparable to that observed in non-transfected cells (see above). When α5 was downregulated, nicotine failed to promote migration at 24 h. In line with our results, previous work by other researchers in the A549 lung cancer cell line could demonstrate that α5-containing receptors also seemed to exert a controlling effect upon nicotine-induced migration [34]. In summary, in MCF7, SW480, and DU145 cells, nicotine promoted proliferation and migration through the α5-subunit-containing nAChRs.

### 3.4. Downregulation of the α5 nAChR Subunit Inhibits PDL1, CD47, and EMT Marker Expression Elicited by Nicotine

Previous reports showed that nicotine is able to induce the epithelial–mesenchymal transition (EMT) [20,70] and that nicotine stimulates the expression of the immunoregulatory proteins CD47 and PDL1 in lung bronchial and epithelial cells [26,29]. Consequently, we next explored whether knock down of the α5 subunit affects the nicotine-elicited expression of these immunoregulatory proteins and EMT markers. Here, we used 10 µM of nicotine to ensure transcript expression of the analyzed RNAs, as it was previously shown that nicotine acts in a dose-dependent manner when analyzing the expression of these proteins [20]. For this test, cells were incubated with 10 μM nicotine for 48 h, and mRNA expression changes in epithelial (E-cadherin) and mesenchymal (vimentin) EMT markers as well as CD47 and PDL1 were analyzed via qPCR. We also used 10 µM MEC to determine if nicotine-induced changes in mRNA expression levels are mediated via nAChRs. As a positive control of nicotine’s effects, we used the lung cancer cell line A549. Figure 6A shows that in A549 cells, nicotine induced the mRNA expression of PDL-1, CD47, and vimentin and the downregulation of E-cadherin, clearly showing the EMT transition. In the MCF7 cell line (Figure 6B), nicotine induced EMT, as previously described [20], but, interestingly, it also upregulated CD47. In DU145 cells (Figure 6C), nicotine increased the mRNA expression of PDL1 but did not do so in SW480 cells (Figure 6D). Nicotine increased the mRNA expression of not only vimentin transcripts but also E-cadherin in SW480 and DU145 cells. When cells were stimulated with nicotine in the presence of MEC, the transcript levels of PDL1, CD47, and both EMT markers returned to their non-stimulated levels (Figure 6), showing that the effect was specific to nAChRs.

Next, we studied how the silencing of α5 affected the nicotine-induced mRNA expression of PDL1, CD47, and both EMT markers in these cell lines. Cells were transfected with α5-siRNA or ns-RNA, and after 24 h, cells were incubated, or not, with 10 μM nicotine for 48 h before analyses. Figure 7 shows that upon downregulation of α5, nicotine did not stimulate the mRNA expression of PDL1, CD47, and EMT markers, indicating that α5 mediates nicotine-induced signaling in the immune regulatory and EMT pathways.

### 3.5. The α5SNP Mutation D398N Increases Basal Proliferation and Migration in the DU145 Cell Line

The results above clearly demonstrate that α5-containing nicotinic receptors are essential for nicotine-induced effects. As previous reports showed that the D398N mutation impairs α5′s function [40,71], we next investigated whether this mutation affects the nicotine-induced effects in proliferation and migration. In this set of experiments, DU145 cells were either transfected with the wild-type α5 construct (α5.pcDNA3.1/V5, α5-WT) or with its mutated version α5D398N (α5D398N.pcDNA3.1/V5, α5-MUT). Both constructs had a V5 tag, and experiments were performed after 24 h. To verify transfection, qPCR and IF determinations were carried out. This cell line was chosen as it had a clear increase in proliferation and migration because of nAChRs activation by nicotine, as opposed to MCF7 and SW480 cells. As shown in Figure 8A, the transfection of α5-WT or α5-MUT increased the expression levels of this α5 subunit in DU145 cells. Figure 8B shows transfection levels via anti-V5 antibodies. We also determined the expression of other nAChR subunits (Appendix A) as a control. Here, no considerable changes in the expression of the other nAChR subunits were detected.

Next, proliferation in the absence or presence of 1 μM nicotine was studied in α5-WT- or α5-MUT-transfected DU145 cells. In α5-WT-transfected cells, 1 µM nicotine increased proliferation compared to non-treated cells (Figure 8C). In α5-MUT-transfected cells, nicotine failed to affect proliferation. Notably, basal proliferation without nicotine was per se higher in cells transfected with α5-MUT than in those transfected with α5-WT (Figure 8C). Migration in cells transfected with the α5-WT construct increased when cells were incubated with 1 μM nicotine (Figure 8D). In cells expressing the mutated α5 subunit D398N, nicotine failed to alter migration. Moreover, basal migration was higher in cells transfected with α5-MUT than in those transfected with α5-WT (Figure 8D).

In lung basal epithelial cells expressing the D398N mutation in α5, basal proliferation is increased through a mechanism involving adenylyl cyclase [32]. Thus, we sought to determine if the increased basal proliferation in DU145 cells expressing α5-MUT is also due to altered adenylyl cyclase activation. In addition, the D398N mutation is localized in a section of the subunit that interacts with heteromeric G proteins [50]. Therefore, we also evaluated if activation of G proteins was involved within the pathway stimulating proliferation. Thus, DU145 cells were transfected with either α5-WT or α5-MUT, and proliferation was detected in the presence or absence of the adenylyl cyclase inhibitor SQ22536, a non-selective inhibitor of AC (SQ, 10 µM), or the G-protein inhibitor pertussis toxin (PTX, 0.1 µM). Our results show that, as described above, the presence of the mutation increased basal proliferation. This increase in basal proliferation was not affected by SQ. Therefore, we reason that AC is not involved in the mechanism (Figure 8E). When PTX was present, the proliferation levels of cells expressing α5-MUT were comparable to those of cells transfected with α5-WT. This result shows that the increased basal proliferation caused by the mutation present in α5 involved a G-protein-mediated mechanism.

Next, we evaluated whether SQ or PTX affected nicotine-induced proliferation in non-transfected DU145 cells (Figure 8F). As shown above, nicotine increased the proliferation of DU145 cells at 24 h and 48 h. The presence of the adenylyl cyclase inhibitor SQ did not affect nicotine-induced proliferation. On the other hand, when PTX was present, nicotine failed to induce proliferation. When given without nicotine, inhibitors did not modify the basal proliferation of DU145 cells (Appendix A). Taken together, our results demonstrate that nicotine induces proliferation through specific activation of the α5 subunit in a G-protein-mediated mechanism.

## 4. Discussion

Ultimately, α5 is expressed in many different types of tissues. However, its physiological and pathophysiological roles are not fully understood. Recent findings have associated this subunit with attention [41], food reward [72], addiction [73], and dopamine release mechanisms [74]. In addition, α5 is associated with nicotine consumption [43] and lung cancer [75]. Elevated α5 protein and mRNA levels are present in different types of human cancers, such as lung, prostate, liver, breast, and gastric cancer [19,27,53,63,64,76,77], and are correlated with smoking status [25,66]. At a cellular level, α5 expression is thought to play a role in nicotine’s regulation of proliferation, migration, and diverse pro-oncogenic signaling pathways in the A549 lung cancer cell line [76], while in other types of cancer cells, its function is not completely understood. In this study, we found that, upon knockdown of α5, nicotine failed to induce proliferation and migration in MCF7, DU145, and SW480 cells. This effect is thought to occur due to loss of the α5 subunit and not due to downregulation of the expression of other subunits, as we detected no change in other nAChR transcripts. Additionally, previous reports have demonstrated that in synaptosomes, neurons, and HEK-transfected cells, the presence of the α5 subunit does not affect the overall membrane expression of the α4β2 or α3β4 receptors [39,71,78,79]. This result suggests that nicotine-induced proliferation is reduced upon α5 knockdown because of impaired receptor function and not because of the aberrant trafficking of receptors. However, further experiments are needed to evaluate receptor traffic in these conditions. Furthermore, no change in basal proliferation or migration was observed when this subunit was absent, suggesting that it is not involved, per se, in these cellular functions. Surprisingly, we did not find α7 to play a role in any nicotine-induced effects, as the specific inhibitor MLA did not change the nicotine-mediated increase in proliferation or migration in MCF7, SW480, and DU145 cells. This result is in contrast with previous observations in lung cancer cell lines such as A549, H1299, PC9, and H1975. In these cell lines, α7 plays an important role in nicotine-induced invasion, migration, proliferation, and EMT [21,23]. Furthermore, in human lung cancer patient samples, this subunit seems to be overexpressed and is associated with smoking levels [19].

Our study also shows that nicotine can induce changes in the mRNA expression of EMT markers and immune regulatory proteins in all cell lines tested, specifically through α5. EMT is a vital step for the facilitation of a malignant and invasive phenotype in cancer, a process characterized by a decrease in epithelial proteins such as E-cadherin, as well as an increase in mesenchymal molecules such as fibronectin and vimentin [20]. Nicotine exposure was previously shown to induce EMT transition in lung and breast cancer cell lines [20,66]. In this study, we demonstrated the contribution of nAChRs in colon and prostate cancer cell EMT transition. Thus, our results are in line with previous observations in lung cancer patients, in which tobacco exposure favors EMT cellular transition [80]. In the human PCa cell line DU145 and the CRC cell line SW480, exposure to nicotine caused an increase in vimentin and the epithelial marker E-cadherin. Previous reports have shown that a loss of expression of the epithelial cell adhesion molecule E-cadherin promotes metastasis and is essential for EMT [81,82]. Other studies have reported that a decrease in this protein is not fully necessary to undergo EMT, nor, consequently, for the pro-oncogenic activity of cancer cells [83,84,85,86]. Nicotine exposure additionally helps maintain immune tolerance and restrain cancer autoimmunity by increasing the mRNA expression of negative immunoregulatory proteins such as PDL1 and CD47. In agreement with previous findings, nicotine increased both protein transcripts in A549 cells and was dependent on nAChR activation. In the MCF7 cell line, only CD47 was increased, and in the DU145 cell line, PDL1 was augmented. However, no effect was observed in the SW480 cell line. Previously, smoking was found to be correlated with increased levels of PDL-1 in lung cancer [25] and CD47 in brain metastases of lung cancer patients [26]. Accordingly, nicotine exposure increased CD47 levels in lung cancer cell lines [26] and PDL1 in lung cancer and melanoma cell lines [27,28]. Furthermore, nicotine-derived nitrosamine ketone (NNK) was shown to induce PDL1 expression in human bronchial epithelial cells [29]. When α5 was silenced in the MCF7, SW480, and DU145 cell lines, nicotine failed to regulate the mRNA expression of all markers evaluated. This result shows that nicotine controls the mRNA expression levels of these proteins via α5-containing nAChRs. Yet additional mechanistic studies are still needed to reveal whether activation of the α5 affects the activity of other signaling pathways/proteins known to be influenced by the expression of such nicotinic receptor subtype, e.g., the vascular endothelial growth factor (VEGF) and epidermal growth factor receptor (EGFR) [33,87].

The non-synonymous single nucleotide polymorphism D398N of the α5 subunit, α5SNP, identified as a candidate mutation for smoking and lung cancer [32,38], was also studied for its contribution to nicotine’s pro-oncogenic effects. We found that overexpression of α5SNP in the prostate cancer cell line DU145 caused an increase in basal proliferation, which was also shown for mouse lung epithelial cells [32]. In addition, overexpression of α5SNP increased migration. Surprisingly, when this mutation was present, nicotine failed to induce its pro-proliferative and pro-migratory effects. This result is aligned with previous observations showing that the presence of this D398N mutation in α3β4α5 nAChRs of HEK293-transfected cells reduced the relative maximum Ca^2+^ response to various nAChR agonists [71]. Moreover, the presence of the D398N mutation in mouse lung epithelial cells increased basal proliferation via a mechanism involving adenylyl cyclase activation [32]. Whilst the nAChRs are very well known to possess ionotropic properties, these receptors also induce metabotropic signaling events [13] and are known to regulate adenylyl cyclase activity [88,89,90]. In the present work, we found that the observed increase in proliferation in DU145 cells expressing the D398N mutation was dependent upon a G-protein-mediated mechanism. Here, when DU145 cells carrying the mutation were in the presence of pertussis toxin, cellular proliferation was halted, whereas incubation with an adenylyl cyclase inhibitor did not intervene with the proliferative activity. The D398N mutation is localized in a section of the subunit that interacts with heteromeric G proteins; more specifically, the intracellular loop region of the α5 subunit interacts with the G_αo_ and G_βγ_ dimers [50]. Interestingly, the amino acid sequence surrounding this mutation was found to be very conserved among species [38], which possibly indicates the presence of an important structural function. Previous findings suggested that the D398N mutation renders the α3β4 receptor more efficient in promoting Ca^2+^ release from intracellular stores [71] and may also play a role in preventing receptor desensitization, thereby maintaining nicotine signaling [40]. In this regard, in neurons and microglial cell lines, α7-mediated Ca^2+^ signals are, in part, mediated by a G-protein-mediated mechanism, ultimately leading to the release of Ca^2+^ from intracellular stores [91,92]. Furthermore, we found that in non-transfected DU145 cells, nicotine increased proliferation through activation of the G-protein pathway via α5. Conversely, a previous report showed that activation of the α4β2 receptors in T-cells led to inhibition of G-protein-mediated signaling, resulting in an increase in proliferation [93]. This result suggests that the contribution of nAChRs in G-protein-mediated signaling depends on cell type and subunit expression patterns. Future studies could provide a more in-depth analysis of the role of nAChRs in G-protein-mediated signaling in this context. Though previous reports have shown that activation of nAChRs has a dual ionotropic-metabotropic functional component, independent of each other [94,95,96,97], further experiments would have to be conducted to determine the involvement of Ca^2+^ permeability or membrane depolarization in our present model.

Whether the endogenous α5 subunit is expressed on intracellular compartments in addition to the plasma membrane in the cell lines studied has not been addressed. In addition, despite the fact that one report has shown its expression in mitochondria [98], there are no data supporting the idea that it has a functional role in that organelle. In contrast, the α7 subunit expressed in mitochondria has been shown to be responsible for regulating cytochrome C release and activating additional signaling pathways important for cell survival [99,100].

Our results point to α5-containing receptors as key signaling nodes of nicotine’s pro-oncogenic effects in diverse cancer types. This discovery is novel since, to date, no other reports have fully determined the importance of α5 in mediating nicotine’s oncogenic functions at the cellular level in breast, colon, or prostate cancer cells. Our findings show the need for α5 inhibitors/modulators to further investigate α5′s role in mediating nicotine’s pro-oncogenic effects in cancer and as a putative drug target. Yet further studies will have to be carried out in order to determine whether the expression of the α5SNP can be used in the future as a tumor-agnostic marker.

## 5. Conclusions

Based on our findings, we show here that the α5 subunit is the key mediator of nicotine’s pro-oncogenic effects, whereas the silencing of the subunit by siRNA rendered the cells studied irresponsive to this stimulus. The signaling events mediated by this subunit are carried through a G- protein mediated mechanism. Most remarkably, the introduction of the α5SNP rendered the cells (i) irresponsive to nicotine, an effect reminiscent of the one observed when the mutation is present in neurons or heterologously expressed receptors in HEK cells [39,40,41,42], and (ii) with an augmented basal proliferation rate, because of increased G-protein activation. This hints that further studies are needed to clarify the role of α5-containing receptors and its SNP in cancer in general, and more specifically the signaling events leading to increased G-protein activation.

## Figures and Tables

**Figure 1 cells-12-02000-f001:**
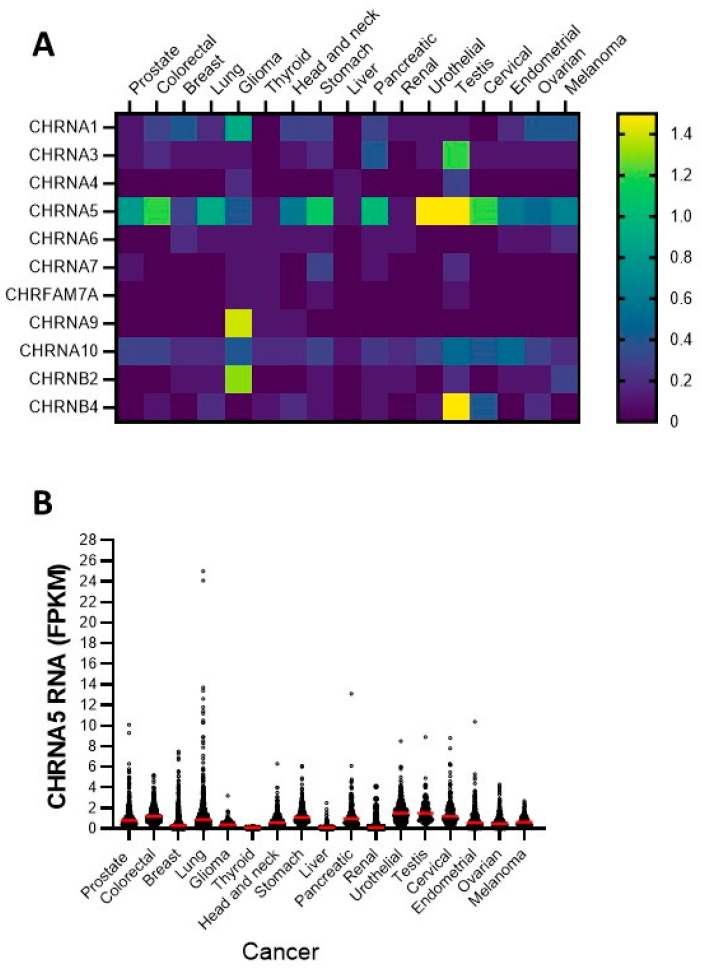
Nicotinic acetylcholine receptor subunit (nAChRs) in different cancer types. Gene *CHRNA1*: α1 subunit; gene *CHRNA3*: α3 subunit; gene *CHRNA4*: α4 subunit; gene *CHRNA5*: α5 subunit; gene *CHRNA6*: α6 subunit; gene *CHRNA7*: α7 subunit; gene *CHRNFAM7A*: dupα7 subunit (partially duplicated α7 subunit isoform); gene *CHRNA9*: α9 subunit; gene *CHRNA10*: α10 subunit; gene *CHRNB2*: β2 subunit; gene *CHRNB4*: β4 subunit. (**A**) Heat map view of several nicotinic acetylcholine receptor (nAChR) subunit genes in human cancers (figure adapted from the Human Protein Atlas, www.proteinatlas.org; accessed on 1 May 2023). The heat map represents the median values per gene for each cancer type. The *CHRNA5* subunit is the most commonly expressed nAChR subunit in all cancer types. (**B**) *CHRNA5* expression levels in different human cancers. Each sample is shown as a circle on the graph, and the median is indicated with a red line. Sequencing data were reported as FPKM (fragments per kilo-base of transcript per million reads mapped). Data were generated by the Cancer Genome Atlas (Pathophysiology of *CHRNA5*, https://www.proteinatlas.org/ENSG00000169684-CHRNA5/pathology; accessed on 1 May 2023).

**Figure 2 cells-12-02000-f002:**
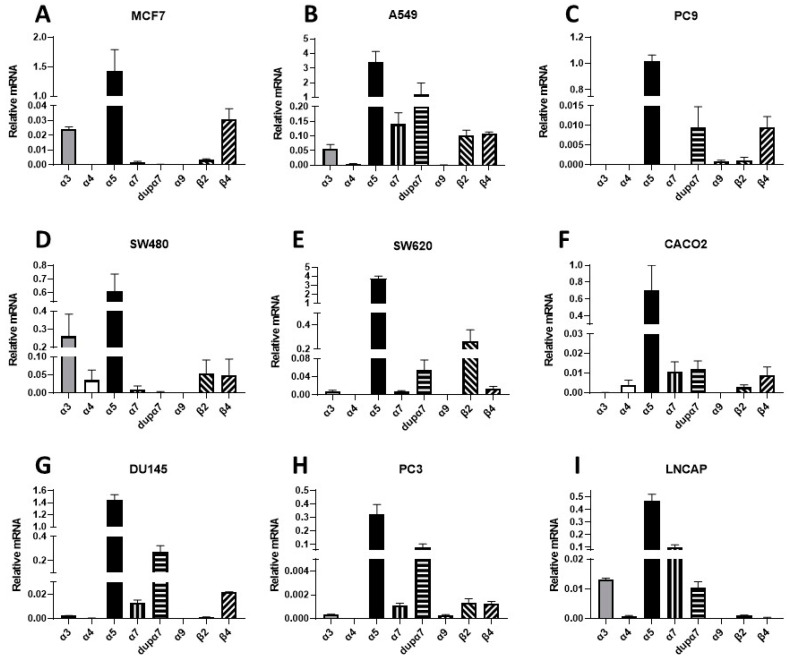
Cell-specific expression patterns of transcripts for nAChR subunits. Human cell lines examined: MCF7 (**A**), breast cancer), A549 (**B**), lung cancer), PC9 (**C**), lung cancer), SW480 (**D**), colorectal cancer), SW620 (**E**), colorectal cancer), CACO2 (**F**), colorectal cancer), DU145 (**G**), prostate cancer), PC3 (**H**), prostate cancer), and LNCAP (**I**), prostate cancer). The gene expression profiles were investigated via qPCR analysis. Bar graphs show the relative expression ± SEM of each subunit normalized to endogenous TATA-binding protein expression. Measurements were made using 3–5 independent experiments performed in triplicate.

**Figure 3 cells-12-02000-f003:**
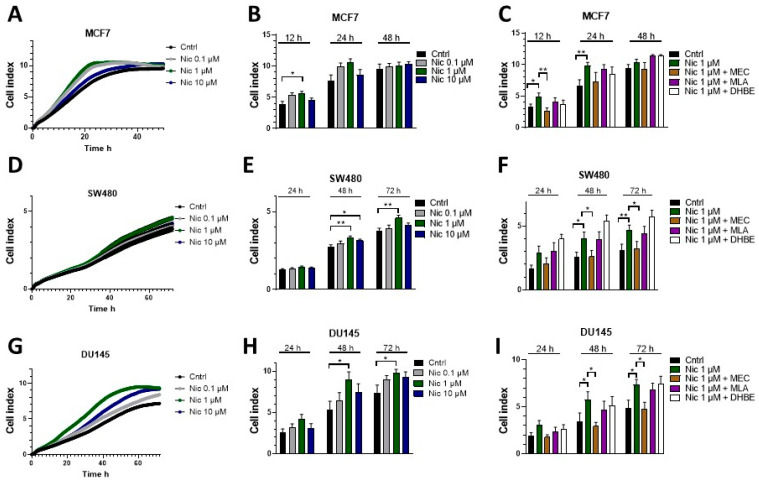
Nicotine increases proliferation through nAChRs in MCF7, SW480, and DU145 human cancer cell lines. Proliferation was evaluated using a label-free impedance-based assay (xCELLigence). (**A**,**D**,**G**) The corresponding cells were treated with 0.1 μM, 1 μM, or 10 μM nicotine. The cell proliferation index is plotted versus time. (**B**,**E**,**H**) Bar diagram of data (mean ± SEM) at the indicated time points from the experiments in A-D-G, correspondingly. (**C**,**F**,**I**) The corresponding cells were treated with 1 μM nicotine in the presence or absence of the inhibitors MEC, MLA, or DHBE. A bar diagram of the proliferation cell index (mean ± SEM) at the indicated time points is shown. Measurements were made using 4–5 independent experiments performed in triplicate. Statistical significance was analyzed using the Kruskal–Wallis test for non-parametric data with the GraphPad Prism (GraphPad 9.1.1 Software) software. A *p*-value of <0.05 was considered significant. * *p* ≤ 0.05, ** *p* ≤ 0.001.

**Figure 4 cells-12-02000-f004:**
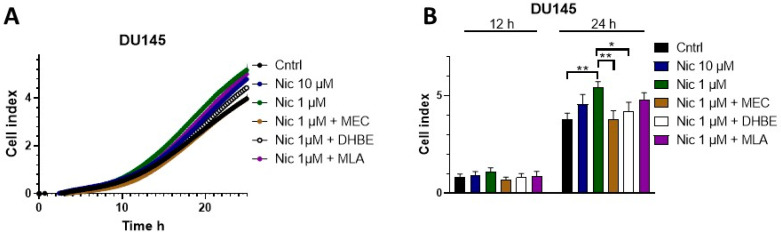
Nicotine increased migration through nAChRs in the human prostate cancer cell line DU145. Analysis was performed using a label-free impedance-based migration assay (xCELLigence). (**A**) DU145 cells were treated with 0.1 μM, 1 μM, or 10 μM nicotine in the presence or absence of the inhibitors MEC, MLA, or DHBE. The cell migration index is plotted versus time. (**B**) Bar diagram of the data (mean ± SEM) at 12 and 24 h from the experiments in A. Measurements were made using 4–5 independent experiments performed in triplicate. Statistical significance was analyzed using the Kruskal–Wallis test for non-parametric data with the GraphPad Prism (GraphPad 9.1.1 Software) software. A *p*-value of <0.05 was considered significant. * *p* ≤ 0.05, ** *p* ≤ 0.001.

**Figure 5 cells-12-02000-f005:**
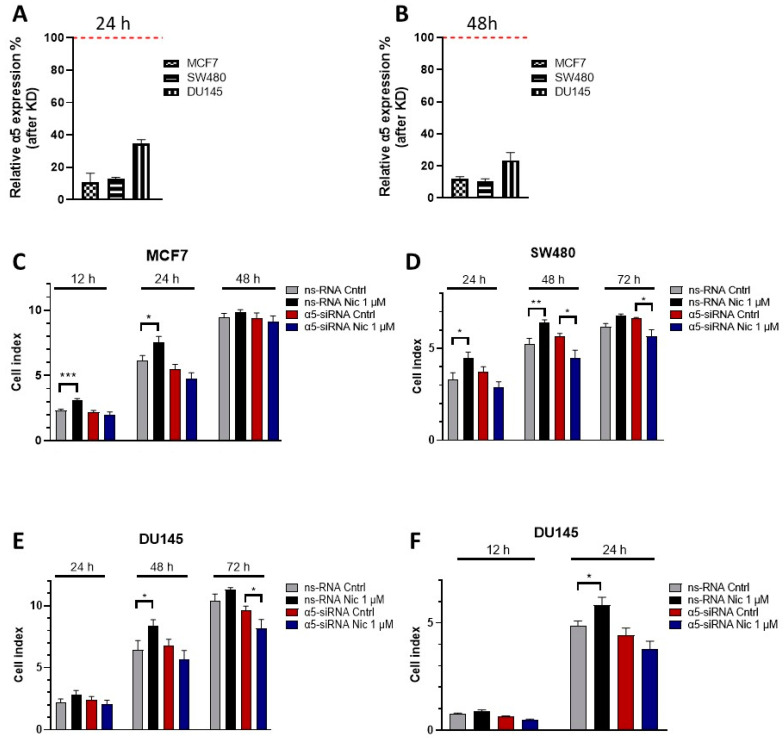
Knockdown of α5 suppressed the proliferative and migratory effects of nicotine on human cancer cell lines. (**A**,**B**) The α5 gene expression demonstrating siRNA knockdown (KD) of the subunit after 24 h (**A**,**B**) 48 h in MCF7, SW480, and DU145 cells. Data are shown as a percentage compared to expression of the subunit in the corresponding cell lines transfected with non-silencing ns-RNA (set to 100% expression, red dashed line). (**C**–**E**) Bar diagram of the cell proliferation index of the indicated cell line when silencing α5 gene expression. α5-siRNA- or ns-RNA-transfected cells were treated with 1 μM nicotine. Data represent the mean ± SEM in the indicated time points. (**F**) Migration cell index of DU145 cells as a result of silencing α5 gene expression. α5-siRNA- or ns-RNA-transfected cells were treated with 1 μM nicotine. The bar diagram shows the mean ± SEM 12 and 24 h after the treatment. Measurements were made using 3 independent experiments performed in triplicate. Statistical significance was analyzed using the Kruskal–Wallis test for non-parametric data with the GraphPad Prism (GraphPad 9.1.1 Software) software. A *p*-value of <0.05 was considered significant. * *p* ≤ 0.05, ** *p* ≤ 0.001.

**Figure 6 cells-12-02000-f006:**
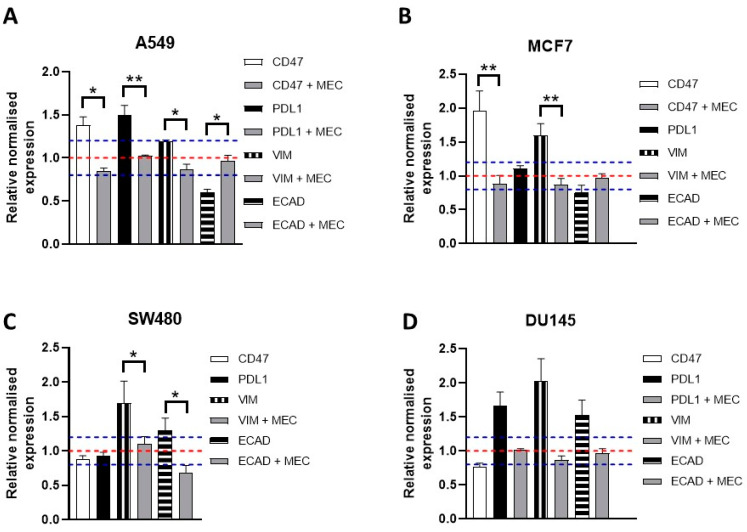
Effects of nicotine on the expression of EMT markers and immunosuppressive proteins in A549 (**A**), MCF7 (**B**), SW480 (**C**), and DU145 (**D**) cancer cell lines. Each figure reflects the expression of CD47, PDL1, VIM (vimentin), and ECAD (E-cadherin) at the mRNA level. The gene expression profile was investigated using qPCR analysis. Bar graphs show the relative expression ± SEM of each subunit normalized to the corresponding expression in non-treated cells (NT; set to a value of 1; red dashed line. Meaningful changes of 1 ± 0.2 is represented by a blue dashed line). Measurements were made using 3–5 independent experiments performed in triplicate. Cells were incubated with nicotine (10 μM) for 48 h. If nicotine had an effect, cells were also incubated in the presence of the inhibitor MEC (10 µM). Statistical significance was analyzed using the Kruskal–Wallis test for non-parametric data with the GraphPad Prism (GraphPad 9.1.1 Software) software. A *p*-value of <0.05 was considered significant. * *p* ≤ 0.05, ** *p* ≤ 0.001.

**Figure 7 cells-12-02000-f007:**
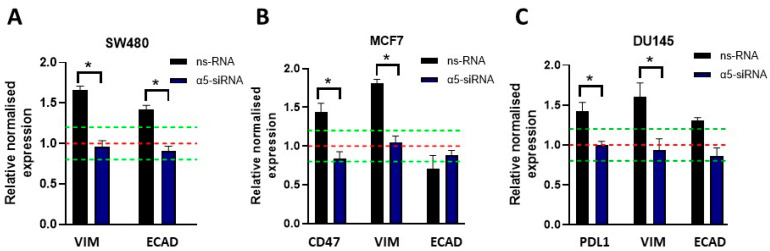
Knockdown of α5 renders cells nonresponsive to nicotine-induced upregulation of EMT markers and immunosuppressive proteins. α5-siRNA- or ns-RNA-transfected cells were treated with 10 μM nicotine for 48 h. The gene expression profiles were investigated using qPCR analysis. Bar graphs show the relative expression ± SEM of each subunit normalized to the corresponding expression in non-treated cells (NT; set to a value of 1; red dashed line. Meaningful changes of 1 ± 0.2 is represented as a green dashed line). Measurements were made using 3–5 independent experiments performed in triplicate. The results obtained for SW480 (**A**), MCF7 (**B**), and DU145 cells (**C**) are shown. Statistical significance was analyzed using the Kruskal–Wallis test for non-parametric data with the GraphPad Prism (GraphPad 9.1.1 Software) software. A *p*-value of <0.05 was considered significant. * *p* ≤ 0.05.

**Figure 8 cells-12-02000-f008:**
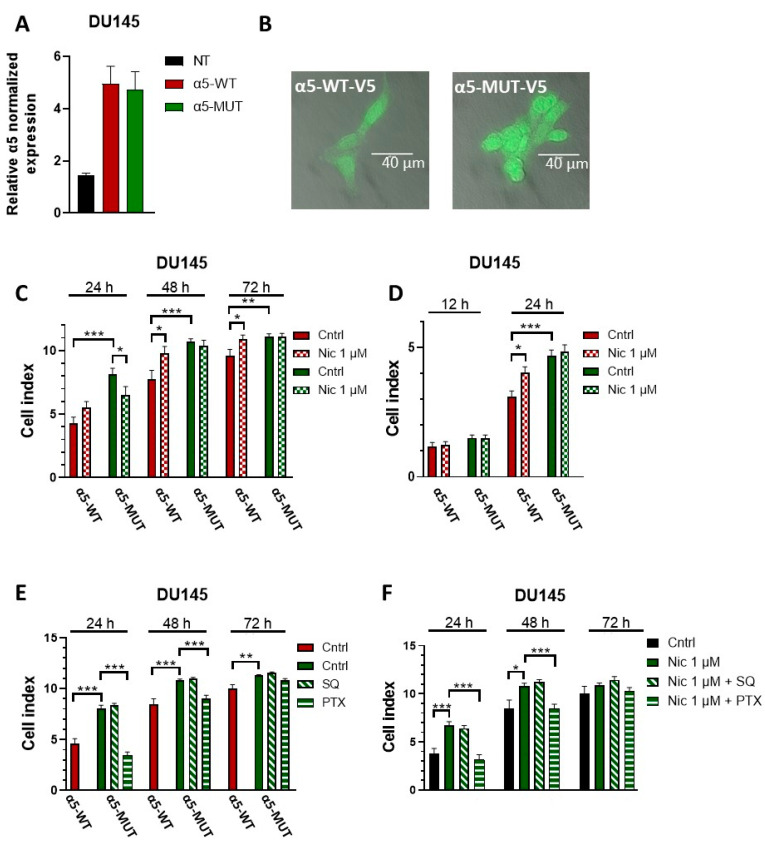
α5SNP enhanced in vitro basal cell proliferation and migration in a prostate cancer cell model. Cells were transfected with either the wild-type α5 construct (α5-WT) or with its mutated version α5D398N (α5-MUT), and functional consequences were studied. (**A**) Depiction of the expression of the α5 subunit in non-transfected DU145 cells (NT; native expression) and in cells transfected with α5-WT (native plus foreign expression) or α5-MUT (native plus foreign expression); mRNA levels were determined via qPCR, and data show the mean ± SEM of three independent cell culture experiments. (**B**) Immunofluorescence images showing the successful overexpression of α5-WT (left) and α5-MUT (right) constructs in DU145 cells; the V5-tagged subunit was detected using incubation with the anti-V5 primary antibody followed by the appropriate Alexa Fluor 488 secondary antibody. Objective 100×, scale bar 40 µm. (**C**,**D**) Cell proliferation index and migration index (respectively) of DU145 cells transfected with α5-WT or α5-MUT-containing plasmids; cells were treated with 1 μM nicotine. The bar diagram shows data (mean ± SEM). (**E**) Cell proliferation index of DU145 cells transfected with α5-WT or α5-MUT-containing plasmids. In the cells transfected with the latter, proliferation was analyzed in the presence or absence of SQ22536 (non-selective inhibitor of ACs, SQ, 10 µM) or PTX (G-protein inhibitor, pertussis toxin, 0.1 µM). The bar diagram shows the data (mean ± SEM). (**F**) Cell proliferation index of non-transfected DU145 cells treated with 1 μM nicotine in the presence or absence of the inhibitors SQ22536 (SQ, 10 µM) or PTX (0.1 µM). Measurements were made using 3 independent experiments performed in triplicate. Statistical significance was analyzed using the Kruskal–Wallis test for non-parametric data with the GraphPad Prism (GraphPad 9.1.1 Software) software. A *p*-value of <0.05 was considered significant. * *p* ≤ 0.05, ** *p* ≤ 0.001, *** *p* ≤ 0.0001.

**Table 1 cells-12-02000-t001:** Proteins and corresponding primers/probes used in the present study.

Protein	Gene Name	Probe ID
α3 subunit *	*CHRNA3*	Hs01088199_m1
α4 subunit *	*CHRNA4*	Hs00181247_m1
α5 subunit *	*CHRNA5*	Hs00181248_m1
α7 subunit *	*CHRNA7*	Hs01063372_m1
dupα7 subunit *	*CHRFAM7A*	Hs04189909_m1
α9 subunit *	*CHRNA9*	Hs00395558_m1
β2 subunit *	*CHRNB2*	Hs01114010_g1
β4 subunit *	*CHRNB4*	Hs00609523_m1
CD47	*CD47*	Hs00179953_m1
PDL-1	*CD274*	Hs00204257_m1
Vimentin	*VIM*	Hs00958111_m1
E-cadherin	*CDH1*	Hs01023895_m1
TATA-binding protein	*TBP*	Hs00427621_m1

* Corresponds to nAChR subunits. All probes are commercially available from ThermoFisher.

**Table 2 cells-12-02000-t002:** Relative expression of nAChR subunits in diverse human cancer cell lines (refers to Figure 2, see above).

nAChR Subunit	MCF7	A549	PC9	SW480	SW620	CACO2	DU145	PC3	LNCAP
α3	0.024 ±0.002	0.057 ±0.014	n.d.	0.261 ±0.054	0.007 ±0.002	n.d.	0.002 ±0.0	n.d.	0.013 ±0.0
α4	n.d.	0.005 ±0.001	n.d.	0.035 ±0.012	n.d.	0.004 ±0.002	n.d.	n.d.	n.d.
α5	1.431 ±0.001	3.396 ±0.743	1.020 ±0.044	0.461 ±0.114	3.739 ±0.284	0.698 ±0.301	1.444 ±0.087	0.321 ±0.075	0.469 ±0.052
α7	0.002 ±0.0	0.141 ±0.033	n.d.	0.010 ±0.004	0.007 ±0.002	0.011 ±0.005	0.013 ±0.002	0.001 ±0.0	0.098 ±0.021
dup α7	n.d.	1.211 ±0.766	0.009 ±0.005	0.002 ±0.001	0.058 ±0.021	0.012 ±0.004	0.271 ±0.053	0.077 ±0.025	0.010 ±0.002
α9	n.d.	0.001 ±0.0	0.001 ±0.0	n.d.	n.d.	n.d.	n.d.	n.d.	n.d.
β2	0.003 ±0.0	0.102 ±0.017	0.002 ±0.0	0.055 ±0.016	0.263 ±0.097	0.003 ±0.001	0.001 ±0.0	0.001 ±0.0	0.001 ±0.0
β4	0.031 ±0.007	0.107 ±0.006	0.009 ±0.003	0.050 ±0.019	0.013 ±0.005	0.009 ±0.004	0.02 2 ± 0.0	0.001 ±0.0	n.d.

Results of the qPCR analysis of nAChR gene expression in the human breast cancer cell line MCF7; the lung cancer cell lines A549 and PC9; the colorectal cancer cell lines SW480, SW620, and CACO2; and the prostate cancer cell lines DU145, PC3, and LNCAP. The data relate to the relative expression ± SEM (normalized to endogenous TATA-binding protein expression) calculated in 3–5 independent experiments performed with samples measured in triplicate; n.d. not determined.

## Data Availability

The data used and analyzed during the current study are available from the corresponding author upon reasonable request.

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
