# Peer review of "Contribution of the α5 nAChR Subunit and α5SNP to Nicotine-Induced Proliferation and Migration of Human Cancer Cells"

_cells, 2023, doi:10.3390/cells12152000_

Round 1

Reviewer 1 Report

This is a novel, well-designed and rigorous study that provides fresh new insight into the role of alpha5 nicotinic receptor subunits in pro-oncogenic actions in various cancers. The authors also show nicely how a polymorphism (D398N) causes a basal increase in proliferation and migration in the DU145 cell line. The experiments are well done, carefully controlled and use of antagonists provide good mechanistic insight. I have only two minor comments, which can easily be addressed in the discussion.

1. Are the effects of alpha5 due to calcium permeate. through the ligand-gated channel or are they a consequence of membrane depolarization (Na+ flux)?

2. There are reports in the literature of nicotinic receptor subunits being expressed on intracellular compartments in addition to the plasma membrane. Are the effects described in the paper a consequence of actions at the plasma membrane?

Reviewer 2 Report

The manuscript submitted by Irida et. al demonstrates the role of a5 receptor in nicotine induced cancer cell proliferation and migration in cancer cell lines from different origin. This role has been reported previously in lung cancer cell lines in a few older publications.  The work is systematic and I appreciate that the author have done relevant experiments for statistical significance.

However, I don’t see a lot of novelty in the manuscript since the data has already been produced in A549 cells by the authors and here they are doing corresponding studies in other cancer cell types and whole study is designed based upon a previous work and the author has limited themselves to it. I see a scope of better manuscript by having an approach which is more comprehensive.

I would suggest the following

Major revisions

11)     Section 3,2- doesn’t explain the time dependence for the results even though most of the results in the figure3 have been done at 3 different time points. Specifically, it needs to be explained why author thinks that 12h result for MCF-7 seems better than 24h and 48h and why it is different amongst the cell lines. Is the reason related to doubling times/ cell growth phases of the cell lines. In this case, I would strongly recommend that these experiments are done on synchronized cells (serum deprivation method) for studying nicotine induced cell proliferation as well as migration.

22)  The results in section 3,5 are quite limited to  two EMT markers and two immune regulatory proteins based upon previous reports. But there are additional reports on crosstalk between α5-nAChR and EGFR signaling in lung cancer ( Low-Dose Nicotine Activates EGFR Signaling via α5-nAChR and Promotes Lung Adenocarcinoma Progression - PMC (nih.gov), I have also seen reports where VEGF signaling is also associated with α5-nAChR in lung cancer. The authors should address this by doing experiments to get correlating results in the additional cell lines. Otherwise, it must be explained why it is not relevant for this manuscript and giving more insight in the discussion section.

33)   Result section is lacking in the choice of cells. It is not explained why the pancreatic, urothelial, stomach, testis, cervical cancer cell lines were not chosen despite having higher levels of CHRNA5 than breast from the human atlas results.  Also the author have not mentioned whether there is an evolutionary significance of the receptor if it is a tumor agnostic marker as this publication suggests.

44)   Authors have also limited themselves to very few techniques, have not done much protein level evaluations( western blots) to produce supporting data. I  recommend to get correlating results for at least EMT marker and CD47, PDL1 expression.

Specific and minor comments

51)   Figure 2E says dup7 instead of dupa7

62)  In lane 285 please mention which cell lines the results of xcelligence assay in the presence of inhibitors has been published

3)  It is not mentioned why the Du145 cells were used for the KI and mutation studies.

4)  Lane 370-71 is unclear,it appears that authors are referring to  their work which is referred, but  author needs to mention that it is their previous work in addition to being more precise so the meaning and  relation to this study is more understandable.

5)     In general, the author must be careful about description in the Figure legends. Figure legend for Figure 5 has a lot of repetition of text which must be avoided. Figure legend for Figure 8 and Figure E, there is mismatch on the time points on the Figure and the one in legends. In addition there is a repetition of text which must be avoided.

6)   There are also minor inconsistencies in Figure8, Size of the heading doenst match on the figures.  

7)   Author should revise the lines 587-592 to make it more easy to comprehend for the readers.

English is fine but authors need to make sure that their discussion section is more comprehensible for the readers

Round 2

Reviewer 2 Report

Thanks for making the changes to the manuscript and adding explanations. I have only the following  suggestions below. 

1) Author has used  'although'  and 'while' a multiple times in the revised sentences added to the manuscript.

fx Line 548-551- this sentence is not comprehensible. Although additional mechanistical studies are needed to reveal whether as in diverse lung cancer cell lines, activation of the α5 affects the activity of other signalling pathways/proteins,  such as the vascular endothelial growth factor (VEGF) and epidermal growth factor re- 551 ceptor (EGFR) [33,87]. 

 same way line 203-206, line 603-605. Please check correct usage of these words and revise sentences accordingly.

2) Author could still avoid repetition of text in Figure 8. 

3) line 603- 604- please use correct english for all your corrections, When the sentence begins with while/ although, it has to have a second supporting  sentence. 

While further studies will have to be carried out to determine whether the expression of the α5SNP can be used as a tumor-agnostic marker.

Could be improved.
